# Unraveling Broadband Near-Infrared Luminescence in Cr^3+^-Doped Ca_3_Y_2_Ge_3_O_12_ Garnets: Insights from First-Principles Analysis

**DOI:** 10.3390/ma17071709

**Published:** 2024-04-08

**Authors:** Wei Zou, Bibo Lou, Mekhrdod S. Kurboniyon, Maksym Buryi, Farhod Rahimi, Alok M. Srivastava, Mikhail G. Brik, Jing Wang, Chonggeng Ma

**Affiliations:** 1School of Science and Optoelectronic Engineering, Chongqing University of Posts and Telecommunications, Chongqing 400065, China; S200602017@stu.cqupt.edu.cn (W.Z.); loubb@cqupt.edu.cn (B.L.); mehrdod-92@mail.ru (M.S.K.); 2Center of Innovative Development of Science and New Technologies, National Academy of Sciences of Tajikistan, Dushanbe 734025, Tajikistan; frahimi2002@mail.ru; 3Institute of Plasma Physics of the Czech Academy of Sciences, U Slovanky 2525/1a, 18200 Prague, Czech Republic; buryi@ipp.cas.cz; 4Current Lighting Solutions LLC., 1099 Ivanhoe Road, Cleveland, OH 44110, USA; srivastaam@outlook.com; 5Centre of Excellence for Photoconversion, Vinča Institute of Nuclear Sciences—National Institute of the Republic of Serbia, University of Belgrade, 11351 Belgrade, Serbia; 6Institute of Physics, University of Tartu, W. Ostwald Str. 1, 50411 Tartu, Estonia; 7Faculty of Science and Technology, Jan Długosz University, Armii Krajowej 13/15, PL-42200 Częstochowa, Poland; 8Academy of Romanian Scientists, 3 Ilfov, 050044 Bucharest, Romania; 9Ministry of Education Key Laboratory of Bioinorganic and Synthetic Chemistry, State Key Laboratory of Optoelectronic Materials and Technologies, School of Chemistry, Sun Yat-Sen University, Guangzhou 510275, China; ceswj@mail.sysu.edu.cn

**Keywords:** Cr^3+^ ions, garnets, near-infrared emission, thermal stability, first-principles calculations

## Abstract

In this study, we conducted an extensive investigation into broadband near-infrared luminescence of Cr^3+^-doped Ca_3_Y_2_Ge_3_O_12_ garnet, employing first-principles calculations within the density functional theory framework. Our initial focus involved determining the site occupancy of Cr^3+^ activator ions, which revealed a pronounced preference for the Y^3+^ sites over the Ca^2+^ and Ge^4+^ sites, as evidenced by the formation energy calculations. Subsequently, the geometric structures of the excited states ^2^E and ^4^T_2_, along with their optical transition energies relative to the ground state ^4^A_2_ in Ca_3_Y_2_Ge_3_O_12_:Cr^3+^, were successfully modeled using the ΔSCF method. Calculation convergence challenges were effectively addressed through the proposed fractional particle occupancy schemes. The constructed host-referred binding energy diagram provided a clear description of the luminescence kinetics process in the garnet, which explained the high quantum efficiency of emission. Furthermore, the accurate prediction of thermal excitation energy yielded insights into the thermal stability of the compound, as illustrated in the calculated configuration coordinate diagram. More importantly, all calculated data were consistently aligned with the experimental results. This research not only advances our understanding of the intricate interplay between geometric and electronic structures, optical properties, and thermal behavior in Cr^3+^-doped garnets but also lays the groundwork for future breakthroughs in the high-throughput design and optimization of luminescent performance and thermal stability in Cr^3+^-doped phosphors.

## 1. Introduction

Among various crystalline solids that are used for optical applications, the compounds with the cubic garnet structure are of special importance and significance. This is a very large family of compounds, whose structure offers numerous opportunities for chemical composition alterations, such as creating solid solutions and/or introducing optically active impurity ions. The garnets can easily accommodate transition metal and rare earth ions, rendering them potential candidates for various optical applications. There has been a recent surge in interest in Cr^3+^-doped garnets due to their broadband near-infrared (NIR) luminescence, with potential applications in medical diagnostics, food analysis, horticultural lighting, night vision, etc. [1,2,3,4,5,6,7]. For instance, Ca_3_Y_2_Ge_3_O_12_: Cr^3+^, synthesized via the solid-state reaction method, exhibits a broadband NIR emission spanning from 700 to 1100 nm, with a peak centered at 800 nm [8]. This emission spectrum aligns perfectly with the absorption frequencies of hydrogen-containing groups *X*-H (where *X*=C, N and O), making it an ideal non-destructive testing tool for food safety applications [9]. Theoretical investigations that provide a comprehensive understanding of the luminescence mechanisms are necessary for enabling the next generation of highly efficient Cr^3+^-activated garnet phosphors.

Extensive systematic spectroscopic analyses have been conducted on garnet crystals doped with Cr^3+^ ions, utilizing the well-established Tanabe–Sugano energy level diagram for 3d^3^ ions in solids [10,11]. Additionally, the exchange charge model within the framework of semi-empirical crystal-field (CF) theory can provide valuable insights into the relationship between the spectroscopic properties of Cr^3+^ ions and their local coordination environments [12]. However, the number of first-principles studies focusing on Cr^3+^-doped garnets within the density functional theory (DFT) framework remains relatively small compared to experimental and semi-empirical theoretical papers on the same topic. This is primarily due to the rather complicated structure of garnets, characterized by a large number of atoms in a unit cell, thus incurring high computational costs. Moreover, most reported DFT calculations on Cr^3+^-doped compounds have predominantly concentrated on ground-state properties, as exemplified by the case of Ca_4_ZrGe_3_O_12_: Cr^3+^ discussed in reference [13]. However, such studies are limiting since they fail to provide the knowledge of Cr^3+^ ions’ ^2^E and ^4^T_2_ excited states, which are important in the design of new useful phosphors. Fortunately, Duan et al. [14,15,16] have successfully applied the ΔSCF-DFT method with non-Aufbau occupations on Kohn–Sham (KS) orbitals to model the excited states ^2^E and ^4^T_2_ of Cr^3+^ ions doped in some oxides. However, such calculations probing Cr^3+^-doped garnets are lacking.

The main goal of the present work is to provide a deeper fundamental understanding of the excited states and the associated luminescence phenomena in Cr^3+^-doped garnets by integrating the first-principles ΔSCF-DFT technique. This integration is essential for addressing the aforementioned gaps in our knowledge. Specifically, we chose the garnet Ca_3_Y_2_Ge_3_O_12_: Cr^3+^ as a case study. In this paper, we conducted an extensive theoretical analysis of its structural, electronic, and optical properties. Special attention was devoted to factors such as the site occupancy, luminescence mechanism, and thermal stability of Cr^3+^ dopants within this garnet matrix. Furthermore, we delved into the challenge of achieving calculation convergence in modeling the excited ^4^T_2_ state of Cr^3+^ ions, employing the approximation of the single-electron configuration t2 2ge1 g, utilizing either the DFT+*U* or hybrid DFT method. Notably, our previous investigation encountered a computational breakdown when describing the excited ^4^T_2_ state of Mn^4+^ ions in K_2_SiF_6_ due to the significant mixing between the constrainedly occupied and unoccupied 3d KS orbitals in the hybrid DFT calculations [17]. Consequently, these predictive calculations and technique development can be readily applied to other systems doped with Cr^3+^ ions, thus offering potential for the high-throughput design of Cr^3+^-doped NIR materials.

This paper is organized as follows: Section 2 contains a description of the calculation method. Section 3 contains all obtained results and their analysis. Finally, the paper is concluded with a summary of our findings.

## 2. Method of Calculations

Our study employed first-principles calculations within the DFT framework, utilizing the Vienna ab initio simulation package (VASP, version 5.4.4.) [18]. Geometric structure relaxations and defect formation calculations were conducted using the Perdew–Burke–Ernzerhof (PBE) functional [19], incorporating an empirical *U* value (*U*_eff_ = 4.0 eV) specifically tailored for the Cr^3+^-3*d* orbitals [14,20]. The electronic structure and optical transitions of both neat and Cr^3+^-doped Ca_3_Y_2_Ge_3_O_12_ were calculated using the hybrid functional of PBE0 with an additional 25% Hartree–Fock exchange [21]. The treatment of semicore electrons for Ca (3s^2^3p^6^4s^2^), Y (4s^2^4p^6^5s^2^4d^1^), Ge (4s^2^3d^10^4p^2^), O (2s^2^2p^4^), and Cr (3p^6^3d^5^4s^1^) was explicitly addressed using the projector augmented-wave pseudopotentials [22,23]. Modeling Cr^3+^ defects in Ca_3_Y_2_Ge_3_O_12_ required a supercell containing 160 atoms, with one Y^3+^/Ca^2+^/Ge^4+^ ion substituted by a Cr^3+^ ion. Sampling the Brillouin zone involved a single *k*-point *Γ* for the total energy and relaxation calculations of the constructed supercell, while a 3 × 3 × 3 *k*-points mesh, based on the Monkhorst–Pack scheme [24], was employed for the host’s unit cell. For both the neat and doped systems, the closed-shell and spin-polarized DFT calculation forms were applied, respectively. A plane-wave basis cutoff energy of 520 eV was employed, with convergence criteria set at 10^−6^ eV for electronic energy minimization and 0.01 eV/Å for Hellman–Feynman forces on each atom.

The formation energy of a defect *X* in the charge state of *q* can be determined as follows [25]: $E^{f}\left(X^{q}\right)=E_{tot}\left[X^{q}\right]-E_{tot}\left[bulk\right]-\sum_{i}n_{i}\mu_{i}+qE_{F}$, where *E*_tot_[*X_q_*] and *E*_tot_[bulk] represent the calculated total energies of the defective and perfect supercells, respectively. The variables *n_i_*, *μ_I_*, and *E*_F_ correspond to the change in the atom number of element *i* (added if *n_i_* > 0 or removed if *n_i_* < 0 with respect to the perfect supercell), the chemical potential of species *i*, and the Fermi energy level, respectively. To account for image charge interaction at periodic boundary conditions and changes in electrostatic potential caused by the defect, the total energies of charged defects were corrected using the method proposed by Durrant et al. [26]. The charge transition level of a defect *X* from its charged states *q* to *q′* (where *q* > *q′*) can be assessed as *ε*(*q*/*q′*) = (*E*_tot_[*X_q′_*] − *E*_tot_[*X_q_*])/(*q* − *q′*) − *E*_VBM_, where *E*_VBM_ represents the energy of the host’s valence band maximum (VBM).

The standard ΔSCF-DFT procedure [27,28] was employed to model the excited states ^2^E and ^4^T_2_ of Cr^3+^ ions in Ca_3_Y_2_Ge_3_O_12_. These states correspond to a spin flip of one t_2g_ electron and a transition of the KS orbital from t_2g_ to e_g_, respectively. Modeling the excited state ^2^E presented no challenges, although the ^4^A_2_-^2^E optical transition energy required adjustment by a scaling factor of 1.5 compared to the DFT-generated value due to the spin contamination effect between the ground ^4^A_2_ and excited ^2^E states [14]. However, it proved to be challenging to represent the excited state ^4^T_2_ in Ca_3_Y_2_Ge_3_O_12_:Cr^3+^. The constrained separation of a pair of electrons and holes to the lowest e_g_ and the highest t_2g_ KS orbitals led to a significant calculation convergence issue in the DFT+*U* and hybrid DFT calculations. This is not surprising, given that the narrow t_2g_-e_g_ energy gap of Cr^3+^ ions in Ca_3_Y_2_Ge_3_O_12_ intensifies the mixing between the lowest occupied e_g_ and the highest unoccupied t_2g_ KS orbitals, which pushes the calculations towards collapse. This problem is exacerbated by the fact that many Cr^3+^-doped garnets with broadband emission are associated with the weak CF case [29]. Considering that the structural disparity between the ground ^4^A_2_ and excited ^4^T_2_ states of Cr^3+^ ions primarily stems from the distinction in the electronic density profiles of the 3d-t_2g_ and e_g_ single-electron states, it is imperative to maintain the single-electron configuration t_2g_^2^e_g_^1^ for modeling the excited state ^4^T_2_ of the Cr^3+^ dopants. However, the two t_2g_ electrons can partially infiltrate into the highest empty t_2g_ KS orbitals to counteract the approach of the lowest occupied e_g_ KS orbital. Therefore, in this study, we proposed two sets of fractional particle occupancy schemes to characterize the geometric structure of the Cr^3+ 4^T_2_ excited state in Ca_3_Y_2_Ge_3_O_12_, as illustrated in Figure 1. In Scheme 1, one of the two t_2g_ electrons is uniformly distributed among the highest two t_2g_ KS orbitals, while the other occupies the lowest t_2g_ KS orbital entirely. In Scheme 2, the allocation of the two t_2g_ electrons is straightforward, with equal distribution among the three t_2g_ KS orbitals.

Slater’s transition-state method [30,31] was employed to estimate the ^4^A_2_-^4^T_2_ excitation and emission energies, respectively, at the equilibrium geometric structures of the ground ^4^A_2_ and excited ^4^T_2_ states. This process involves examining the disparities in energy between the lowest e_g_ and highest t_2g_ KS orbitals in the density of states (DOS) diagrams obtained from such calculations based on the single-electron configuration t_2g_^2.5^e_g_^0.5^, as depicted in Figure 1. The associated zero-phonon line (ZPL) energy can be readily determined by applying the Franck–Condon principle. Additionally, the Stokes shift energy can be calculated by evaluating the difference between the excitation and emission energies of the corresponding optical transitions.

## 3. Results and Discussion

### 3.1. Ground States of Both Neat and Cr^3+^-Doped Ca_3_Y_2_Ge_3_O_12_

#### 3.1.1. Structural Properties and Defect Site Occupancy

Ca_3_Y_2_Ge_3_O_12_ crystallizes in the conventional cubic garnet structure, with the $Ia\bar{3}d$ space group and an experimental lattice constant of 12.8059 Å [32]. Within this crystalline framework, the coordination environments of the constituent cations manifest intriguing symmetries and spatial arrangements, as shown in Figure 2. Specifically, the Y^3+^ ions occupy octahedral coordination sites, characterized by point group symmetry *S*_6_, wherein each Y^3+^ ion is coordinated with six O^2−^ ions at an equidistant Y-O distance of 2.234 Å. In contrast, the Ge^4+^ ions reside in tetrahedral sites, displaying point group symmetry *S*_4_, with a coordinated arrangement of four O^2−^ ions at an identical distance of 1.766 Å. Meanwhile, the Ca^2+^ ions are found within dodecahedral coordination environments, distinguished by point group symmetry *D*_2_, each surrounded by eight neighboring O^2−^ ions. Notably, such a coordination environment results in two distinct Ca-O distances (2.469 and 2.560 Å).

The calculated structural data for the Ca_3_Y_2_Ge_3_O_12_ host, encompassing lattice constants, internal anion position, unit cell volume, and bond lengths of Y^3+^-O^2−^, Ge^4+^-O^2−^ and Ca^2+^-O^2−^, demonstrate substantial agreement with the earlier-discussed experimental descriptions, as outlined in Table 1. The observed slight overestimation is ascribed to the inherent characteristics of the generalized gradient approximation employed in the PBE functional. Considering the ionic radius difference between Cr^3+^ dopants and the three substitutional sites available [33], it is anticipated that the introduction of Cr^3+^ at the Y^3+^ and Ca^2+^ sites will induce a contraction in their local coordination environments, while the opposite effect is expected at the Ge^4+^ sites. The Cr^3+^-O^2−^ bond lengths and the unit cell volume changes upon Cr^3+^ doping at the three cationic sites, calculated using the PBE+*U* method, strongly corroborate this empirical conclusion, as evidenced by the data comparisons presented in Table 1. The experimentally refined unit cell volume after Cr^3+^ doping tends to decrease compared to the host case (refer to Figure 2c in the reference [8]). This, combined with the calculated findings, suggests a preference for Cr^3+^ dopants to substitute at Y^3+^ and Ca^2+^ sites over the Ge^4+^ sites. A further inference can be drawn, indicating that the Y^3+^ sites are more accommodating to Cr^3+^ ions than the Ca^2+^ sites, owing to the closer alignment of ionic radii between Y^3+^ and Cr^3+^ ions within a six-coordinated-ligand environment.

To conclusively determine the preferential site occupancy of Cr^3+^ ions in Ca_3_Y_2_Ge_3_O_12_, we calculated the formation energies of Cr dopants at the three cationic sites within the PBE+*U* framework. In this study, the chemical potential for oxygen atoms was established by considering a gas of O_2_ molecules, expressed as $\mu_{O}=\frac{1}{2}E_{O_{2}\left[gas\right]}+\Delta \mu_{O}.$ Here, $E_{O_{2}\left[gas\right]}$ represents the calculated total energy per formula unit for O_2_ gas, and Δ*μ*_O_ is related to the contribution arising from gas partial pressure (*P*) and sintering temperature (*T*). Under the specified experimental conditions (*T* = 1450 °C and *P =* 1 atm [8]), Δ*μ*_O_ was determined as −2.051 eV, following the formula expression provided in the reference [34]. The chemical potentials of other atoms (Ca, Y, Ge, and Cr) were straightforwardly derived from the calculated total energies per formula unit of their respective bulk binary oxides, based on the obtained oxygen chemical potential. These are determined by the following equations: $\mu_{Ca}=E_{CaO\left[bulk\right]}-\mu_{O}$, $\mu_{Y}=1/2\;(E_{Y_{2}O_{3}\left[bulk\right]}-3\mu_{O})$, $\mu_{Ge}=E_{GeO_{2}[bulk]}-2\mu_{O}$, and $\mu_{Cr}=1/2\;(E_{{Cr}_{2}O_{3}\left[bulk\right]}-3\mu_{O})$. Figure 3 illustrates the formation energies of Cr ions substituting at the three cationic sites as a function of Fermi energy. Inspection of Figure 3 reveals that the charge state of Cr ions located at the Ge^4+^ sites undergoes a transition from “+4” to “+3” as the Fermi energy increases. This aligns with the common understanding in coordination chemistry, where transition metal ions with a 3d^3^ electronic configuration tend to be oxidized at a tetrahedral site in the absence of additional constraints from physics or chemistry. In contrast, those occupying the Ca^2+^ and Y^3+^ sites consistently maintain a “+3” charge state. Simultaneously, the formation energy of Cr^3+^ ions substituting the Y^3+^ sites consistently remains lower than those in the Ca^2+^ and Ge^4+^ sites. Consequently, defects involving Cr ions substituting at the Y^3+^ sites dominate, and the charge state of Cr ions is predominantly “+3” in Ca_3_Y_2_Ge_3_O_12_. This theoretical fact is fully confirmed by the experimental XRD analysis reported previously [8] and aligns with the calculated structural properties of Ca_3_Y_2_Ge_3_O_12_: Cr^3+^ discussed above. Hereafter, if not specifically emphasized, we exclusively focus on the case wherein Cr^3+^ ions occupy the Y^3+^ sites in Ca_3_Y_2_Ge_3_O_12_ for the description of the structural, electronic, and optical properties of Ca_3_Y_2_Ge_3_O_12_: Cr^3+^.

Despite the results obtained by Cui et al. [9], only the results for a single Cr^3+^ location in the host (at the Y^3+^ site) are shown here, which is based on the Cr^3+^ preference to occupy the octahedral sites in crystalline solids (see Figure 3).

#### 3.1.2. Electronic Properties

The band structure, along with the DOSs, were computed for pristine Ca_3_Y_2_Ge_3_O_12_ utilizing the PBE0 functional, taking into account the optimized geometric structure of the host, as illustrated in Figure 4. The calculated band gap displays a direct character and measures 5.82 eV, marking a significant improvement compared to the result of 3.32 eV obtained with the PBE functional. This closely aligns with the experimentally determined optical band gap of the host (5.71 eV), determined through the Kubelka–Munk function and the Tauc relation applied to the measured diffuse reflection spectra [8]. The top of the valence bands (VBs) appears relatively flat, similar to other oxygen-based garnets [35], while the bottom of the conduction bands (CBs) exhibits notable dispersion, with a single CB dipping down at the *Γ* point. This observation strongly suggests high electron mobility in the CBs and the localization behavior of holes in the VBs. The calculated DOS diagrams provide further insight into the composition of the band edges. The VBs’ top is predominantly influenced by the O-2*p* orbitals, whereas the CBs’ bottom is primarily composed of the Ca-3*d*, Y-4*d*, Ge-4*s*, and O-2*sp* orbitals.

Figure 5 depicts the DOS diagrams of Cr^3+^-doped Ca_3_Y_2_Ge_3_O_12_, obtained from the PBE0 calculations using the optimized geometric structure when Cr^3+^ ions occupy the Y^3+^ sites. As anticipated, new states associated with the Cr-3d orbitals emerge within the band gap. The lower Cr-3d-t_2g_ KS orbitals with spin up are observed to subtly split into two bands, positioned slightly above the top of the VBs. Meanwhile, the higher Cr-3d-e_g_ KS orbitals with spin up, localized in the CBs, remain as one band without any observable splitting. The calculated results fully align with the fundament knowledge in group theory: the triply degenerate t_2g_ transforms into a single-fold *A* and a doubly degenerate *E*, whereas the doubly degenerate e_g_ is maintained as *E* when Cr^3+^ ions occupy the Y^3+^ octahedral sites with the point group symmetry *S*_6_ [36]. All the Cr-3d KS orbitals with spin down are deeply buried into the CBs.

### 3.2. Excited States ^2^E and ^4^T_2_ of Cr^3+^-Doped Ca_3_Y_2_Ge_3_O_12_

#### 3.2.1. Structural Properties

The equilibrium geometric structures of the excited states ^2^E and ^4^T_2_ of Cr^3+^ ions in Ca_3_Y_2_Ge_3_O_12_ were determined using the ΔSCF technique with the PBE+*U* method, as illustrated in Figure 6. In the case of the excited state ^2^E, the calculated local coordination environment of Cr^3+^ ions maintains the initial point group symmetry of *S*_6_, with a slight bond length contraction of 0.007 Å in comparison to its ground state ^4^A_2_. This minor change is anticipated since the ^2^E state is not associated with an orbital change but rather a spin flip when compared to ^4^A_2_. In contrast, the optimized equilibrium geometric structures of the excited state ^4^T_2_, utilizing two sets of fractional particle occupancy schemes tailored for the calculation convergence issues mentioned in the computational methodology section, undergo a significant Jahn–Teller distortion, resulting in a symmetry descent from *S*_6_ to its subgroup *C_i_*. This distortion involves a notable axial expansion (at least an increase of 0.15 Å) and a slight equatorial compression in the [CrO_6_]^9−^ complex. And the Cr^3+^-O^2−^ bond lengths, initially identical, split into three groups. These calculation results align with semi-empirical CF analyses on the Jahn–Teller effect for 3d ions in solids [37]. Additionally, we considered the volume of the [CrO_6_]^9−^ complex as an index to characterize the distortion level of the excited state ^4^T_2_ with respect to the ground state ^4^A_2_. The calculated results of 11.6742 (^4^A_2_), 12.3496 (Scheme 1 for ^4^T_2_), and 12.5107 (Scheme 2 for ^4^T_2_) Å^3^ on this parameter indicate a potential over-relaxation risk in the equilibrium geometric structure obtained from the second fractional particle occupancy scheme, or an under-relaxation case for Scheme 1 (which will be discussed later).

#### 3.2.2. Optical Properties and Luminescence Mechanism

Upon analysis of the obtained geometric structures for the ground state ^4^A_2_ and excited states ^2^E and ^4^T_2_ of Cr^3+^ in Ca_3_Y_2_Ge_3_O_12_, we conducted the PBE0 calculations to determine the excitation, emission, ZPL, and Stokes shift energies for the optical transitions ^4^A_2_-^2^E and ^4^A_2_-^4^T_2_, employing the ΔSCF technique within Slater’s transition-state method. The resulting values, along with the available experimental data, are tabulated in Table 2. The observed comparison between the ZPL energies of the excited states ^2^E and ^4^T_2_ and the ground state ^4^A_2_ indicates a weak CF case in Ca_3_Y_2_Ge_3_O_12_: Cr^3+^. Consequently, the experimentally observed broad NIR emission should be attributed to the ^4^A_2_-^4^T_2_ optical transition, not solely due to its spin-allowed transition nature. Remarkably, the calculated excitation, emission, and Stokes shift energies for the ^4^A_2_-^4^T_2_ optical transition demonstrate excellent agreement with the experimental values, particularly when predicated on the ^4^T_2_ geometric structure optimized with the first fractional particle occupancy scheme, as opposed to Scheme 2. An over-relaxation phenomenon is discerned in the ^4^T_2_ geometric structure optimized by Scheme 2, manifesting through a markedly larger calculated Stokes shift energy. The superiority of Scheme 1 becomes apparent, as the chosen particle occupations on the three t_2g_ KS orbitals align aptly with their energy distribution corresponding to the combination of a single-fold *A* and a doubly degenerate *E*. Furthermore, the ^4^A_2_-^2^E optical transition displays a negligible configuration coordinate change, evidenced by the calculated Stokes shift energy of 0.02 eV. This observation, in conjunction with the significantly larger Stokes shift energy of 0.27 eV observed in the optical transition of ^4^A_2_-^4^T_2_, receives robust support from the structural data presented in the previous section.

To understand the luminescence mechanism of the materials under investigation, we constructed a host-referred binding energy (HRBE) diagram for Cr^3+^-doped Ca_3_Y_2_Ge_3_O_12_ following Dorenbos’s standardization [38], as illustrated in Figure 7. The energy level position of the ground state ^4^A_2_ of Cr dopants within the band gap was determined by employing the charge transition level *ε*(+1/0) (denoted as *ε*(Cr^4+^/Cr^3+^)), derived from the PBE0 total energy calculations. Simultaneously, the positions of the excited states ^2^E, ^4^T_2_, and ^4^T_1_ were ascertained by considering their calculated ZPL energies relative to the ground state ^4^A_2_. It is worth noting that the estimation of the ^4^A_2_-^4^T_1_ ZPL energy involves the energy difference between the two experimentally observed ^4^A_2_-^4^T_2_ and ^4^A_2_-^4^T_1_ excitation energies (i.e., ~0.72 eV taken from reference [8]). Inspection of Figure 7 reveals three distinct excitation pathways that induce the luminescence: host absorption from VB to CB and the ^4^A_2_ → ^4^T_1_ and ^4^A_2_ → ^4^T_2_ transitions of Cr^3+^ dopants. Evidently, the excitation efficiency of the host absorption is the lowest, given the considerable separation of the ground ^4^A_2_ and luminescent ^4^T_2_ energy levels from the top of the VBs and the bottom of the CBs, respectively. The isolated nature of the luminescence, free from the interference of the host’s electronic structure, also ensures potentially excellent quantum efficiency for applications in NIR light sources. Both anticipated observations are substantiated by the experiments (refer to Figure 4 and see Section 3.4 in reference [8]).

#### 3.2.3. Thermal Stability

The luminescent energy level observed in the investigated compound has been attributed to the excited state ^4^T_2_ of Cr^3+^ dopants. Consequently, the thermal quenching effect in Cr^3+^-doped Ca_3_Y_2_Ge_3_O_12_ arises primarily from the thermally activated crossover between the potential surfaces of the ground ^4^A_2_ and excited ^4^T_2_ states [39]. It is crucial to note that the crossover between the potential surfaces of the ground ^4^A_2_ and excited ^2^E states appears challenging due to a nearly negligible change in configuration coordinates relative to the ground state ^4^A_2_. The thermal excitation energy (*E_a_*) of Ca_3_Y_2_Ge_3_O_12_: Cr^3+^, a pivotal parameter for describing the thermal stability of materials, can be defined as the energy difference between the crossover point of the potential surfaces of the ground ^4^A_2_ and excited ^4^T_2_ states and the equilibrium structure point of the excited state ^4^T_2_. To evaluate *E_a_*, we constructed a configuration coordinate diagram of Cr^3+^ ions in Ca_3_Y_2_Ge_3_O_12_. This involved considering the equilibrium structure points of the ground ^4^A_2_ and excited ^4^T_2_ states (denoted as *Q_g_* and *Q_e_*, respectively), utilizing the calculated excitation, emission, and ZPL energies of the optical transition ^4^A_2_-^4^T_2_ (denoted as *E_x_*., *E_m_*. and *E_ZPL_*, respectively), and applying a one-dimensional harmonic approximation for the potential surfaces of the ground ^4^A_2_ and excited ^4^T_2_ states, as illustrated in Figure 8.

By utilizing the calculated average Cr^3+^-O^2−^ bond lengths of the [CrO_6_]^9−^ complex in the ground ^4^A_2_ and excited ^4^T_2_ states as the horizon coordinate values of the *Q_g_* and *Q_e_* points, respectively, the crossover point (denoted as *Q_T_*) between the potential surfaces of the ground ^4^A_2_ and excited ^4^T_2_ states can be ascertained. Consequently, the thermal excitation energy *E_a_* was determined to be 0.266 eV. This value closely aligns with the reported thermal excitation energy of 0.25 eV, derived by fitting a modified Arrhenius equation to the measured temperature dependence of the emission intensity of Ca_3_Y_2_Ge_3_O_12_: Cr^3+^ [8]. This good agreement between the calculated and experimentally estimated thermal barrier values serves as solid proof of the validity of the performed analysis and allows for a further investigation of the role of non-radiative processes in the deactivation of the excited electronic states. Such a prediction concerning thermal excitation energy holds significant value for the smart search for novel NIR Cr^3+^-doped phosphors with high thermal stability. Additionally, the energy differences between the two *Q_g_* and *Q_e_* points on the potential surfaces of the ground ^4^A_2_ and excited ^4^T_2_ states were assessed, yielding values of 0.21 and 0.06 eV, respectively (the sum of these values corresponds to the Stokes shift energy). This indicates that the primary energy loss during the luminescence kinetics process takes place in the ground-state relaxation following emission.

The reliable outcomes of the present paper validate the robustness of the fractional particle occupancy scheme developed in this study, effectively overcoming the calculation convergence challenges present in the DFT+*U* and hybrid DFT modeling of the excited ^4^T_2_ state of 3d^3^ ions in solids. Additionally, this scheme can serve as a complement to an alternative approach quite recently proposed by Duan et al., which involves deactivating the 3d subspace diagonalization to address the same encountered problem [16]. Beyond enriching our fundamental understanding, this study establishes a foundation for future endeavors in the high-throughput design of novel Cr^3+^-doped phosphors, placing emphasis on both high thermal stability and the luminescent properties required for NIR applications.

## 4. Conclusions

In conclusion, our thorough investigation, employing first-principles calculations within the DFT framework, has successfully unraveled broadband NIR luminescence in Cr^3+^-doped Ca_3_Y_2_Ge_3_O_12_ garnets. This comprehensive exploration has provided valuable insights into the intricate interplay among geometric and electronic structures, optical properties, and thermal behavior. The findings are summarized below:The results from both the structural analysis and the defect formation energy calculations indicate a tendency for Cr^3+^ dopants to preferentially occupy Y^3+^ sites rather than Ca^2+^ and Ge^4+^ sites. Comparing the optimized geometric structure of the ground state ^4^A_2_ of Cr^3+^ ions, the excited state ^4^T_2_ exhibits a significant Jahn–Teller distortion, characterized by a notable axial expansion and a slight equatorial compression in the [CrO_6_]^9−^ complex. In contrast, the excited state ^2^E primarily retains the initial ground-state structure, undergoing a negligible change.The host material Ca_3_Y_2_Ge_3_O_12_ features a direct band gap of 5.82 eV, allowing sufficient space to accommodate the multiple energy levels of Cr^3+^ dopants. The calculated positions of the ground ^4^A_2_ and excited ^4^T_2_ energy levels within the band gap underscore the isolated nature of Cr^3+^ optical centers from the host’s electronic structure. This discovery further supports the observed higher quantum efficiency.The calculated energies for the excitation, emission, and Stokes shift associated with the optical transitions ^4^A_2_-^2^E and ^4^A_2_-^4^T_2_ show a much better agreement with the experimental values. The energy comparison of the optical transitions ^4^A_2_-^2^E and ^4^A_2_-^4^T_2_ indicates that Cr^3+^ ions are located in a weak CF. The identification of three distinct excitation pathways that induce the ^4^T_2_→^4^A_2_ luminescence suggests that the excitations of Cr^3+^ ions to the ^4^T_1_ and ^4^T_2_ states are more efficient.Our accurate prediction of thermal excitation energy has paved a direct path to providing fundamental analysis of the thermal quenching process in phosphors doped with 3d^3^ ions, using the configuration coordinate diagram.

## Figures and Tables

**Figure 1 materials-17-01709-f001:**
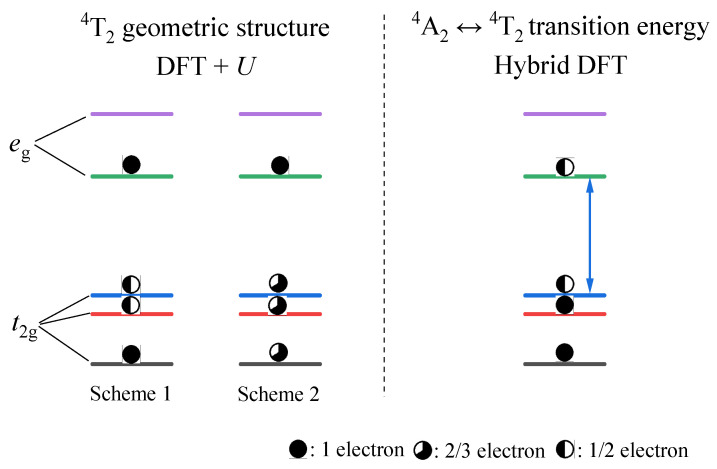
Schematic diagrams depicting the fractional particle occupancy schemes employed to determine the geometric structure of the ^4^T_2_ excited state and the ^4^A_2_-^4^T_2_ optical transition energies of Cr^3+^ ions located in an octahedral environment. The left and the right parts are referred to in the text as Scheme 1 and Scheme 2, respectively.

**Figure 2 materials-17-01709-f002:**
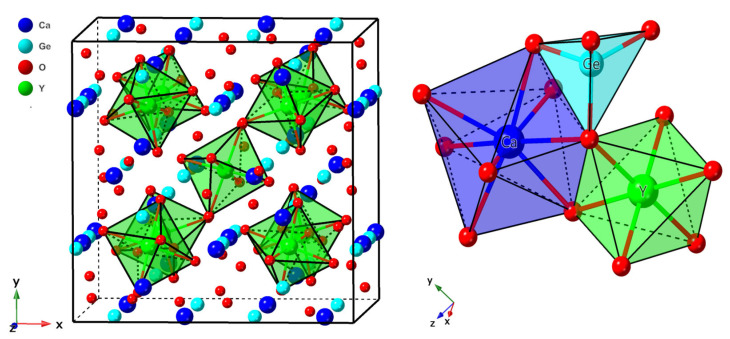
Schematic representations of the crystal structure of Ca_3_Y_2_Ge_3_O_12_, illustrating the spatial arrangement of the constituent cations and their corresponding local coordination environments.

**Figure 3 materials-17-01709-f003:**
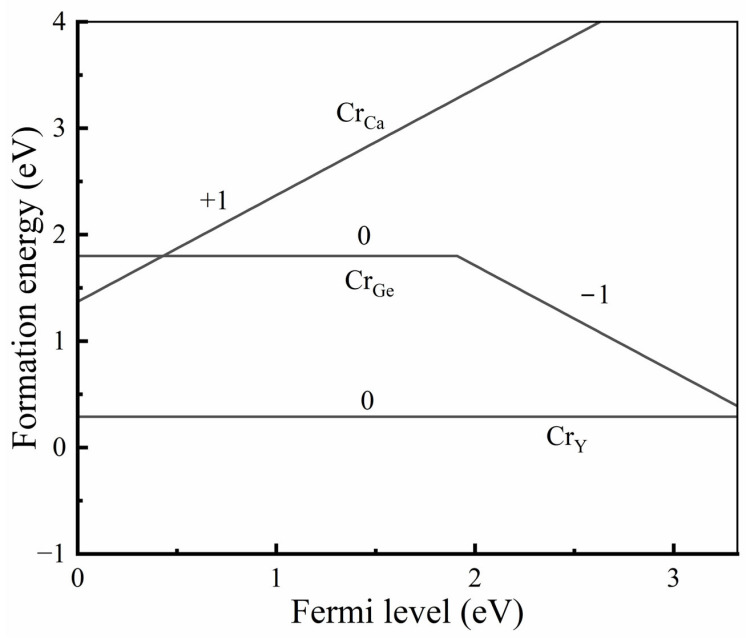
Calculated formation energies of Cr substitutions (Cr_Y_, Cr_Ca_, and Cr_Ge_) in Ca_3_Y_2_Ge_3_O_12_ plotted against Fermi energy. The VBM energy is referenced to zero, and the integer values on the line segments represent the total charges of the analyzed defective systems.

**Figure 4 materials-17-01709-f004:**
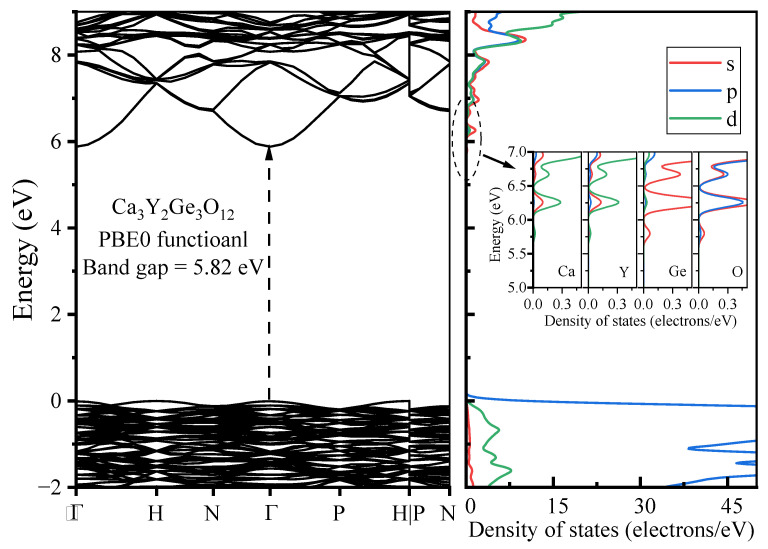
Calculated band structure and DOS diagrams for pristine Ca_3_Y_2_Ge_3_O_12_. The insets display the projected DOS diagrams focused on the CBs’ bottom. The VBM energy is referenced to zero. The symbols *Γ*, *H*, *N*, and *P* denote the high-symmetry *k*-points (0 0 0), (1/2 − 1/2 1/2), (0 0 1/2), and (1/4 1/4 1/4), respectively.

**Figure 5 materials-17-01709-f005:**
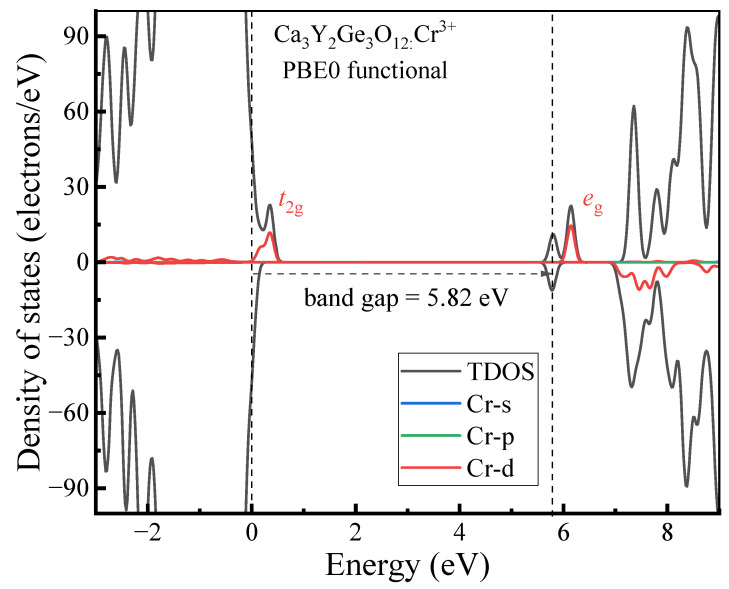
Calculated density of states diagrams of Cr^3+^-doped Ca_3_Y_2_Ge_3_O_12_ in the ground state ^4^A_2_. The valence band maximum energy is referenced to zero.

**Figure 6 materials-17-01709-f006:**
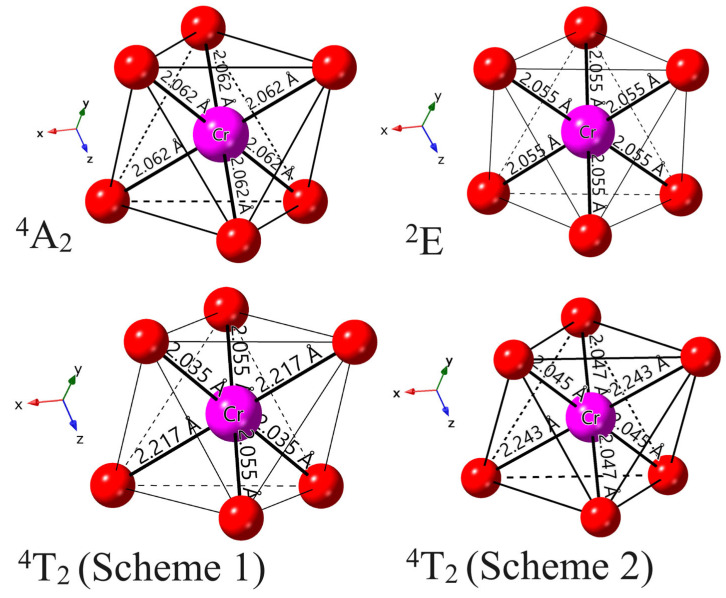
Schematic representations of the local coordination environments of Cr^3+^ dopants in the ground state ^4^A_2_ and the excited states ^2^E and ^4^T_2_, including the results obtained using two sets of fractional particle occupancy schemes.

**Figure 7 materials-17-01709-f007:**
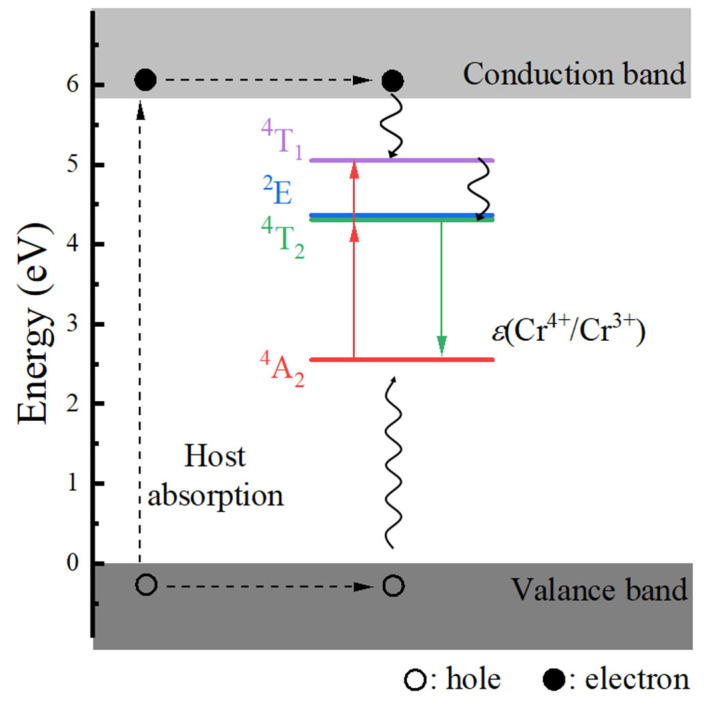
Calculated host-referred binding energy diagram of Cr^3+^-doped Ca_3_Y_2_Ge_3_O_12._ The notation *ε*(Cr^4+^/Cr^3+^) represents the calculated charge transition level *ε*(+1/0). The details to determine the energy level positions of the ground state ^4^A_2_ and the excited states ^2^E, ^4^T_2_, and ^4^T_1_ can be found in the text.

**Figure 8 materials-17-01709-f008:**
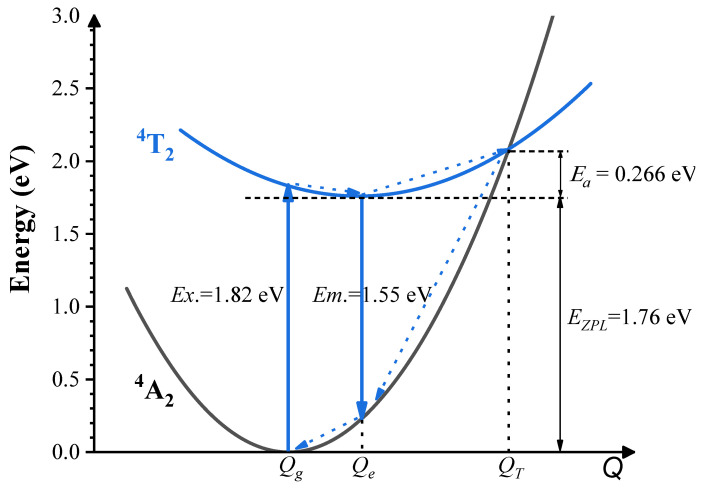
Schematic depiction of the calculated configuration coordinate diagram of Cr^3+^ ions in Ca_3_Y_2_Ge_3_O_12_. The ground state ^4^A_2_ energy is used as the reference point (zero). *E_x_*., *E_m_*., *E_ZPL_* and *E_a_* represent the calculated excitation, emission, and zero-phonon line energies of the optical transition ^4^A_2_-^4^T_2_, and the thermal excitation energy, respectively. *Q_g_*, *Q_e_*, and *Q_T_* denote the equilibrium structure points of the ground ^4^A_2_ and excited ^4^T_2_ states, along with the potential surface crossover point between the two states, respectively. The blue dotted arrows indicate the non-radiative transitions. Excitation, emission, and zero-phonon line energies (1.82 eV, 1.55 eV, 1.76 eV) correspond to the wavelengths of 681 nm, 800 nm, and 704 nm.

**Table 1 materials-17-01709-t001:** Comparison of the calculated and experimental structural properties of both neat and Cr^3+^-doped Ca_3_Y_2_Ge_3_O_12_ in their ground states: lattice constants (*a = b = c*, in Å), non-dimensional coordinates of internal anion position (*x*, *y*, *z*), unit cell volume *V* before and after Cr^3+^ doping (in Å^3^), and bond lengths of Y^3+^-O^2−^, Ge^4+^-O^2−^ and Ca^2+^-O^2−^ in the pure host, along with Cr^3+^-O^2−^ bond lengths upon Cr^3+^ doping at the three cationic sites (in Å).

System	Parameter	Calc.	Expt. *^a^*
Ca_3_Y_2_Ge_3_O_12_	*a = b = c*	12.9381	12.8059
	O (*x*, *y*, *z*)	0.9644, 0.0557, 0.1604	0.9637, 0.0567, 0.1609
	*V*(host)	2165.7508	2100.0533
	Y^3+^-6O^2−^	2.245	2.234
	Ge^4+^-4O^2−^	1.789	1.766
	Ca^2+^-4O(1)^2−^	2.486	2.469
	Ca^2+^-4O(2)^2−^	2.596	2.560
Ca_3_Y_2_Ge_3_O_12_:Cr^3+^	*V*(Cr^3+^/Y^3+^)	2155.4599	-
	Cr^3+^/Y^3+^-6O^2−^	2.062	-
	*V*(Cr^3+^/Ge^4+^)	2187.9207	-
	Cr^3+^/Ge^4+^-4O^2−^	1.932	-
	*V*(Cr^3+^/Ca^2+^)	2148.4651	-
	Cr^3+^/Ca^2+^-4O(1)^2−^	2.142	-
	Cr^3+^/Ca^2+^-4O(2)^2−^	2.576	-

Note: *^a^* Ref. [32].

**Table 2 materials-17-01709-t002:** Comparison of the calculated and experimental excitation, emission, ZPL, and Stokes shift energies of the optical transitions between the ^4^A_2_ ground state and the excited states ^2^E and ^4^T_2_ of Cr^3+^ in Ca_3_Y_2_Ge_3_O_12_ (all in eV).

	Excitation	Emission	ZPL	Stokes Shift
^2^E	1.80	1.78	1.79	0.02
^4^T_2_				
Scheme 1	1.82	1.55	1.76	0.27
Scheme 2	1.82	1.39	1.71	0.43
Expt. *^a^*	1.83	1.55	-	0.28

Note: *^a^* Ref. [8].

## Data Availability

Data are contained within the article and can be obtained from the authors upon request.

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
