# Peer review of "Unraveling Broadband Near-Infrared Luminescence in Cr3+-Doped Ca3Y2Ge3O12 Garnets: Insights from First-Principles Analysis"

_materials, 2024, doi:10.3390/ma17071709_

Round 1

Reviewer 1 Report

Comments and Suggestions for Authors

Referee report on:

„Unraveling broadband near-infrared luminescence in Cr3+- 2 doped Ca3Y2Ge3O12 garnets: Insights from first-principles analysis”

by

Wei Zou, Bibo Lou, Mekhrdod S. Kurboniyon, Maksym Buryi, Farhod Rahimi, Alok M. Srivastava, Mikhail  G. Brik, Jing Wang, Chong-Geng Ma

In this publication, the authors, using first-principles calculations within the framework of functional density theory, tried to calculate the probability of Cr3+ being placed in the Y3+ site and two other positions, i.e., replacing with Ca2+ and Ge4+ in Ca3Y2Ge3O12 garnet as well as the energy bandgap and other spectroscopic properties.

The whole paper is original and relevant to the field. The calculations were performed for the first time for this material. There is no need of improvements regarding the methodology.

In the article by Cui et al. cited in the manuscript as [9], it is stated that Cr3+ enters two sites replacing Ca2+ and Y3+. Two peaks were found there in the emission spectra assigned to these two Cr3+ positions and two emission decay profiles were adequate for each site.

Experimental results are always the final and best verification of theoretical results. Therefore, the authors should write at least that "despite the results obtained by Cui et al., they can present only the results for a single Cr3+ location in the host which replace with Y3+." This is the minimum required, but it would be great if the authors could provide additional results calculated for the Ca site.

The references in this article are appropriate. And the paper is very good and could be published after minor revisions. No additional comments.

Author Response

We thank the reviewer for a positive evaluation of the manuscript. The sentence suggested by the reviewer was added in this form: “Despite the results obtained by Cui et al. [9], only the results for a single Cr3+ location in the host (at the Y3+ site) are shown here, which is based on the Cr3+ preference to occupy octahedral sites (see Figure 3). “ We would like to stress that the total energy calculations (Fig. 3 of the manuscript) indicate that the formation energy of Cr3+ at Ca2+ site is significantly higher, so we would not expect such mechanism of substitution.

Reviewer 2 Report

Comments and Suggestions for Authors

Dear Author:

In my viewpoint, the manuscript number materials-2927512 titled "Unraveling broadband near-infrared luminescence in Cr3+-doped Ca3Y2Ge3O12 garnets: Insights from first-principles analysis" can be accepted to publication after major revision.

See, I doesn't see the theoretical emission spectrum in the manuscript, thus the aspect "broad" of the emission wasn't observed by reader.

In this sense, energy levels positioned in the band gap, below of Fermi level isn't approached.

Further mention to radiative and non-radiative transition should be added

Comments on the Quality of English Language

The English  level has been revised by professional revisor, however seem that isn't a researcher.

Author Response

We thank the reviewer for a positive evaluation of the manuscript. The sentence “This good agreement between the calculated and experimentally estimated thermal barrier values serves as a solid proof of validity of the performed analysis and allows for a further investigation of the role of non-radiative processes in the deactivation of the excited electronic states.” Was added when discussing the agreement between the calculated ant experimentally deduced thermal barriers.

The suggested method of calculations does not allow to calculate the spectrum itself; we can only locate the impurity ion electronic states in the host band gap, what we have done and shown in Fig. 5.

Reviewer 3 Report

Comments and Suggestions for Authors

The manuscript reports the investigation of the broadband near-infrared luminescence in the Cr3+-doped Ca3Y2Ge3O12 garnet by employing first-principles calculations within the density functional theory framework.

The work is relevant to contemporary efforts focused on developing novel brad NIR phosphors. It uses experimental data reported in the literature. The Theoretical DFT framework is typical with emphasis on the reliable outcomes obtained from the fractional particle occupancy scheme developed. It would be interesting to apply this approach to other Cr-doped garnets to test their reliability and accuracy.

It would be interesting to a reader if the authors consider and refer to the ESR data confirming the Cr4+/Cr3+ transition in Ca3Y2Ge3O12 or other similar garnets.

What is the optimal and maximum Cr doping level to Ca3Y2Ge3O12 garnet?

The authors should consider/discuss the Cr-Cr pair interaction when doping the Ca3Y2Ge3O12 garnet when doping with a high Cr concentration.

What would be the expected intervalence change transfer of Cr-Cr pairs and clusters?

What is the probability of Cr-Cr dimers, trimers, and higher-order cluster formation in Ca3Y2Ge3O12 garnet? How this will affect the luminescence properties, especially at a high Cr doping level?     

Comments on the Quality of English Language

 Minor editing of English language required.

Author Response

We thank the reviewer for a positive evaluation of the manuscript. The suggested by the reviewer analysis of the Cr cluster formation is an interesting problem, but given the different cluster agglomerations it would imply time-consuming calculations performed for large supercells. It was out of scope for the present paper; we can consider this topic for the future publications and we hope that the reviewer would accept our reply. We also plan to perform the ESR experiments for these samples in the future. As for the optimal Cr level in this garnet – according to Ref. [9] of the manuscript, it is about 2 %.

Round 2

Reviewer 2 Report

Comments and Suggestions for Authors

Dear Author:

In my viewpoint, the manuscript number  titled “Unraveling broadband near-infrared luminescence in Cr3+ doped Ca3Y2Ge3O12 garnets: Insights from first-principles analysis” need major revision prior to publication acceptance.

I would like comments a set of topics that compose items of the major revision.

As a whole comment, the manuscript was submitted to periodic in the materials area, then all finds should be adapted to materials area, instead spectroscopic area.

Also, seems that vicious of language emerge at some points. See,Item Conclusion: "In conclusion, our thorough investigation...", See, Title, abstract, Conclusion are domain of "contributions", then should be new scientifics contributions.

The English need further revision, see as example, only in the “Abstract” item:

-          There are a great number of adverbs, in any language adverbs don’t added information to text, then in addition delete all adverbs along manuscript, only in the Abstract, six adverbs, as follow: an extensive, successfully,  effectively,  excellent, Importantly,  consistently ,   

With relation to scientific English:

-          Now: …Cr3+ dopants; change to …Cr3+ cations    

Title item:

At moment, the broadband extension, and its formation is unclear, the text is segmented and there are a great number of acronyms.  Also, it is necessary to check meaning and meaningfulness, if the broadband is a set of emissions very close, where is modeling, discussion and diagramming.  
Abstract Item:

I think that in the phrase "Furthermore, the accurate prediction of thermal excitation
energy yielded valuable insights into the thermal stability..." contain erroneous aspects.
Thermal stability is correlated with low expansion or low shrinkage, and increase in
the temperature of fusion point. I suggest that authors verify this find.

Results and Discussion item:

All legends of Figures should be inspected, in this work all acronyms should be changed by its words, as example: ZPL is Zero Phonon Line

Discussion and legend of Figure 7 needs further attention. At this point, should be presented to readers further correlation between this plot, that is based on discrete levels and broadband. Excitations and decays should be commented in the text, in according symbols used in the Figure, arrows dashed are unclears, wave arrow and color of arrows should be explained. I suggest that in addition to eV units, also units are wavelength be added.

Conclusion item:

As major comment, the text of this item is too long and should be shortened.

Also, the structure is strange and out of standard since contain references and mention to Figure. See, I recommend that part that contain reference be added to previous text, in its specific position of discussion. With relation to mention to a Figure of manuscript is necessary rewrite this part

Also, there is a long text that isn’t scientific conclusion, then shouldn’t be added to topic.

Therefore, it is necessary to delete the part below noticed:

 “The reliable outcomes obtained above validate the robustness of the fractional particle occupancy scheme developed in this study, effectively overcoming the calculation convergence challenges present in the DFT+U and hybrid DFT modelling of the excited 4T2  state of 3d3 ions in solids. Additionally, this scheme can serve as a complement to an alternative approach quite recently proposed by Duan et al., which involves deactivating the 3d subspace diagonalization for addressing the same encountered problem [16]. Beyond enriching our fundamental understanding, this study establishes a foundation for future  endeavors in the high-throughput design of novel Cr3+-doped phosphors, placing emphasis on both high thermal stability and the luminescent properties required for NIR applications.”

Comments on the Quality of English Language

The English level requires further revision
